# Are we preparing healthy & responsible adolescents? Exploratory qualitative study to understand the health and social issues of adolescent living in Karachi, Pakistan

**Narjis Rizvi** [ID]*, **Sarah Saleem, Jawaria Mukhtar Ahmed, Sayyeda Ezra Reza, Rawshan Jabeen, Saleem Jessani**

Community Health Science CHS, Aga Khan University, Karachi, Pakistan

* narjis.rizvi@aku.edu

**Data Availability Statement:** The data is only available upon request given institutional privacy

## Abstract

This study aimed to understand the lives of adolescents living in squatter settlements of Karachi, Pakistan regarding their assigned roles and responsibilities, health and social issues, their decision-making process, and mechanism and channels of information. An exploratory descriptive qualitative study design was employed. Purposive sampling techniques were used to carry out Focus Group Discussions (FGDs) with adolescents (n = 10, Participants = 190), adolescents' parents (n = 10, Participants = 180), and In-Depth Interviews of with key adolescent stakeholders (n = 20). Adolescent stakeholder mapping was conducted for enrolling participants. The data were analysed thematically using deductive and inductive approaches. Based on gender norms parents assign specific roles and responsibilities to adolescent girls and boys. Due to societal norms, communication gap exists between adolescents and their parents. The most popular information channels are social media platforms and friends. Adolescents reported being subjected to a variety of physical, sexual, mental, social, and environmental pressures. Lack of guidance from parents, inadequate knowledge and skills to deal with physical, sexual, mental, social and environmental hazards, and misuse of social media lead to risky decisions, injuries, and social instability. This study underlines the urgent need for targeted interventions for addressing gender issues and improving adolescents' decision-making and life skills. We recommend Behavior Change Communication interventions to dismantle gender stereotypes and support a balanced domestic environment for children's education and well-being, awareness raising among parents, teachers, and healthcare providers about adolescent risks emphasizing their role in youth guidance, and advocacy for youth-led forums to co-create educational content engaging parents, educators, and health professionals focusing on life skills. These strategies would turn demographic transitions into productive dividends.

and ethical restrictions on sharing it publicly. According to our institutional privacy policy and research ethics, we are not able to publicly release our qualitative data as it contains potentially identifiable and confidential patient information. Reasonable data access requests can be considered by contacting Aga Khan University Ethical Review Committee (erc.pakistan@aku.edu).

**Funding:** This work was supported by the Bill & Melinda Gates Foundation, Aman Foundation and the David & Lucile Packard Foundation (Grant SGA 0614-01 awarded to N.R.). The funders provided technical support in the study design and supervised in data collection for ensuring the data quality. However, they had no role in data analysis and manuscript preparation. All the authors who are involved in this research and manuscript development were employees of Aga Khan University.

**Competing interests:** The authors have declared that no competing interests exist.

## Background

The World Health Organization (WHO) defines adolescence as being between the ages of 10 and 19 years [1] while the Lancet Commission reports an inclusive age range of 10–24 years [2]. Regardless of age, adolescence is the transition period between childhood and adulthood [2]. During this phase, adolescents perform and experience many tasks for the first time including the acquisition of the emotional and cognitive abilities for independence, completion of education, initiation of employment, social engagement, taking responsibilities, formation of lifelong relationships, choosing a spouse, marriage, sexual debut, beginning of a family, and parenthood. As a result, this transition provides numerous opportunities for physical, social, and emotional growth [3]. However, it also poses several risks to adolescent health and well-being because this is the stage at which gender differences crystallize and shape their roles [3]. Therefore, adolescents must be equipped with the necessary knowledge and skills to promote positive development and effectively manage changing needs, roles, and responsibilities[4].

Adolescents account for 16% of the global population, with 22% living in South-East Asia Region (SEAR). Adolescents have been overlooked in global health and social policy until recently, which is one reason they have seen fewer health gains and economic development than other age groups. Universal Health Coverage (UHC), with the goal of "health for all at all ages," provides a significant medium for governments to meet adolescents' health needs and improve their overall development and well-being [1]. The Global Strategy for Women's, Children's and Adolescents' Health presents an excellent opportunity for investment in adolescent health and well-being, with the vision of creating a world in which women, children, and adolescents have the right to health, well-being, development, and the ability to contribute to the development of sustainable societies by 2030 [5]. Nonetheless, achieving the Sustainable Development Goals (SDGs) and targets will require large-scale investments in developing the capabilities of adolescents and young adults [6].

Adolescents account for 22.7% of the population in Pakistan [7, 8]. Surveys and research primarily focus on married adolescents and fertility issues [9]. Only one population-based survey on unmarried adolescents, focusing on menarche/puberty, has been conducted on unmarried adolescents that focused on menarche/puberty [10] The scant literature on unmarried adolescents reveals that either no knowledge is provided to assist them in managing their new roles and responsibilities [11]or that the knowledge provided is insufficient, inappropriate, and delayed [12]. Research also shows that unmarried adolescents, despite their willingness, are forbidden to share and seek advice about their reproductive, mental, and social problems [13] as these are societal taboos [14, 15]. Furthermore, societal norms forbid family elders from discussing sexual health, particularly with unmarried adolescents [16]. Adolescents face barriers for accessing health and social care when they need it due to restrictive legislative frameworks, out-of-pocket service costs, stigma, and community attitudes [6] In the absence of adequate knowledge and timely guidance, adolescents make risky decisions that often have negative consequences for their personality, social interactions, and health, [17]. Furthermore, they have been dealing with a variety of health, sexual, reproductive, and social issues [13], which cause confusion and stress [18]. This emphasizes the importance of developing Sexual and Reproductive Health and Rights (SRHR) policies that are relevant to adolescents' health and social issues and, needs, and transformation of these policies into effective programs [19]. Hence, it is critical to understand the health and social needs of Pakistani adolescents.

In Karachi, Pakistan, squatter settlements lack basic amenities such as health, education, sports, and other social services and infrastructure which are essential for adolescents' normal growth and upbringing. This study aimed to understand adolescents' health and social issues,

roles, responsibilities, their decision-making process, and mechanism and channels of information in Karachi, Pakistan.

## Methods

### Study design & purpose

A qualitative descriptive study design was employed to understand the lives of adolescents through exploring adolescents, parents and stakeholders' perspectives by using purposive sampling technique.

### Study setting and participants

- This study was conducted as part of the Sukh Initiative, a collaboration between the Aman, the Bill & Melinda Gates, and the David & Lucile Packard Foundations. This project was implemented by a consortium of six national and international organizations including the Community Health Sciences (CHS) Department of Aga Khan University (AKU). The primary objective of 'Sukh' was to increase contraceptive use by 15% among Karachi, Sindh, Pakistan's 100 million underserved peri-urban population. Additionally, the project also explored the lives of adolescents.

- The study was conducted in ten squatter settlements in four towns of Karachi where the 'Sukh' field-based health and social centers were established (Refer to Table 1).

- Participants included, a diverse set of key stakeholders—adolescents aged 16 to 18, adolescent's parents, government sports and youth affairs officers, political leaders, social activists, Non-Governmental Organization (NGO) administrators, educational leaders from public and private sectors, healthcare professionals, and community health workers. *List of Participants and data collection techniques (Refer to Table 2)*

- A purposive sampling technique, detailed in Table 1, was designed to capture a wide range of perspectives based on gender, geographic location, and social roles, thus providing a comprehensive understanding of the multifaceted adolescent experience in these communities. This diverse participant pool offered varied viewpoints, enriching our study with insights from different socioeconomic, cultural, and professional backgrounds.

**Table 1. Town and station-wise list of selected areas.**

| Town | Area | Station |
|------|------|---------|
| Bin Qasim Town | Shah Latif Town Sector 19/20 | 1 |
| | Dabla Para Rehri Goth | 2 |
| | Umer Marvi Goth | 3 |
| Korangi | Bhittai Colony | 4 |
| | Ittehad Colony | 5 |
| | Area I Korangi No. 5 | 6 |
| | Sector 33 | 7 |
| Landhi | Future Colony | 8 |
| | Mansehra Colony | 9 |
| Malir | Ghazi Town | 10 |

**Table 2. List of participants and data collection techniques.**

| Data Collection Techniques | Participants' Categories | Numbers of FGDs/IDIs |
|---|---|---|
| Focus group discussion | Adolescents | 10 (Total number of Participants = 190) |
| In-depth interviews | Adolescents' Parents | 10 (Total number of Participants = 180) |
| Key Informant Interviews | Sports & Youth Affairs Officer<br>Local Political Leader<br>Community Leaders<br>Youth NGO Administrators<br>Head of Schools (Public & Private)<br>Head of Madrassas<br>Healthcare Provider<br>Pharmacist<br>Lady Health Worker | 20 |

- Ethical approval for this study was obtained from the Aga Khan University Ethical Review Committee (AKU-ERC). Written informed consent was provided by all study participants. Informed consent included permission to audio record the interviews and use anonymized quotes.

## Data collection

Using semi-structured guides, Focus Group Discussions (FGDs) with adolescents (16–18 years) and parents, and In-Depth Interviews (IDIs) with key stakeholders were conducted to collect data. The FGD guide was pilot-tested with six participants in one FGD, while the interview guide was pilot-tested with five participants. Both guides included open-ended questions and probes to elucidate perspectives on adolescents' characteristics, knowledge of pubertal changes, activities they have been involved in, health and social issues they face, their decision-making process, and mechanisms and channels accessed by them to gather information. During discussions, care was taken to avoid stifling dialogue however opportunity was provided for the emergence of new domains.

Data were collected iteratively by researchers of the Sukh project between 1st December 2014 to 31st January 2017 at local community centers for FGDs and offices of participants for IDIs),. To establish rapport, participants were invited in a cordial manner to voluntarily participate in the interview with full assurances of anonymity, privacy, and confidentiality. The parents of the adolescents then gave their informed, written consent for voluntary participation. For adolescents, parental consent was obtained before taking adolescents ascent.

The duration of FGDs and IDIs was 60 minutes and 40 minutes, respectively. All sessions were audio recorded, pseudo-anonymized, and transcribed. The field notes also included participant impressions and emerging topics. Data collection continued until data saturation was reached and no new information was yielded. All data were encrypted, stored, and processed in accordance with applicable data protection regulations. There was no financial reward for participants. The quality of the research was ensured by using COREQ Checklist (Consolidated criteria for Reporting Qualitative research).

## Data analysis

Using inductive approaches, the data was analyzed and coded and sub-themes and themes were developed [20, 21]. Findings were clarified using the one-sheet-of-paper method [22] and codes were assigned to participants: (FGD = Focus Group Discussion participant; KI = Key Informant participant; F = Female; M = Male) and study sites (SS = Squatter Settlement, I-X). The findings of FGDs with adolescents were triangulated with FGDs with adolescents' parents

and KIs with stakeholders. The findings were translated into English while preserving their culturally specific meanings by using QSR-Nvivo 12 software.

## Study rigor

Rigor was maintained throughout the research by ensuring credibility and confirmability [23, 24]. For credibility, member checking was performed which included reconfirming the FGD content with participants. Secondly, peer debriefing was also performed where the researchers shared and discussed the research findings, interpretations, and methodology with peers. To maintain confirmability, the researchers indulged in reflexivity to ensure mitigation of personal views on research process and ensuring adequate audit trail by keeping comprehensive records of the research process, including rough notes and interview transcriptions.

## Results

In total, 10 Focus Group Discussion with adolescents (n = 190 participants) and adolescent's parents (n = 180) were conducted respectively. In FGDs with adolescents 53% (n = 105,) were females while 44.7% (n = 85,) were males. In FGDs with adolescent's parents, 55.5% (n = 100,) were mothers while 44.5% were fathers(n = 80,). A total of twenty stakeholders were interviewed.

The *Town and Station-Wise List of Selected Areas is given in* Table 1.

Data Collection Techniques, Participants Categories and the Number of FGDs/IDIS is given for example the Fig 1 shows Characteristics of Parents (FGD participants n = 180) and Fig 2 shows: Characteristics of adolescents (FGD participants n = 190)

### 1. Adolescents characteristics

Approximately all the adolescents, their parents, and stakeholders believe that the community members are well aware that adolescence begins between the ages of 12 to 14 for girls and 15

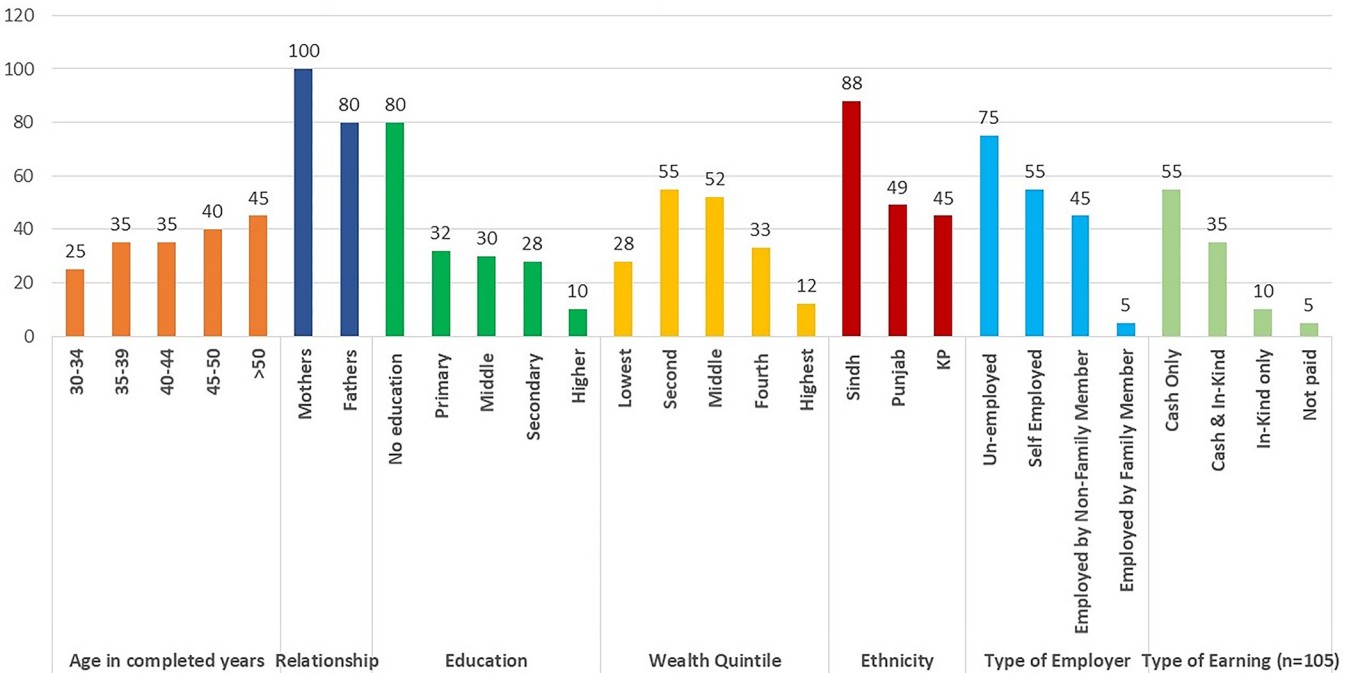

**Fig 1. Characteristics of parents (FGD participants n = 180).**

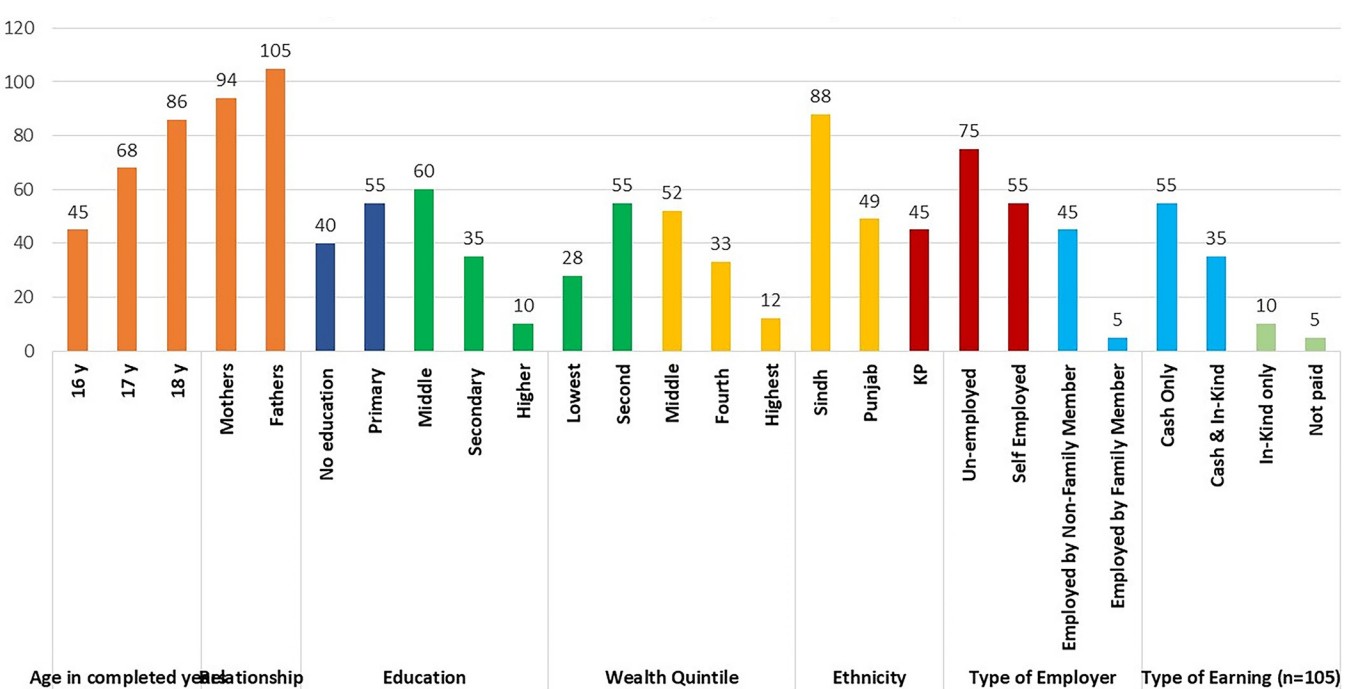

**Fig 2. Characteristics demographic of adolescents (FGD participants n = 190).**

to 18 for boys. According to all the respondents, the indicators of the beginning of adolescence are menstruation and breast enlargement in girls and the appearance of mustache, beard, and voice changes in boys. They mentioned:

> "Adolescence begins between the ages of 12 and 14 for girls and 15 and 18 for boys and is marked by physical changes."- *(Adolescents, Parents & Stakeholders)*

## 2. Knowledge about pubertal changes

According to the findings adolescents lack knowledge of pubertal physical changes and are unprepared to manage them. Parents do not inform adolescents about puberty, its associated changes, and the necessary preparations to manage them. Most of the participants reported that because adolescents lack knowledge about pubertal changes, they perceive them as health problems.

> "Girls have no prior knowledge of the menstrual cycle; sudden bleeding confuses them, and they believe it is the result of an illness or internal injury."—*(Female Adolescents, Slum I, V & VIII & Lady Health Worker)*.

Adolescent males reported that societal norms prohibit adolescents from discussing their pubertal changes and sexual issues and getting guidance.

> "Boys cannot discuss their pubertal changes and sexual concerns with anyone as this is unacceptable."–*(Male Adolescents, SS II, VI & IX & Pharmacist)*

The majority of adolescents of both genders reported that coping with these pubertal changes alone negatively impacts their mental and emotional health.

> "We manage our reproductive and sexual issues ourselves, and this give us emotional and mental stress."–*(Male & Female Adolescents: Slum-I, III, VIII)*

### 3. Adolescent's role and responsibilities

The participants unanimously reported that based on gender norms parents assign different roles and responsibilities to girls and boys. Girls' mobility is restricted, they are expected to stay at home and help their mothers with household chores while in a few low-income families' girls begin paid employment in factories or as domestic servants. Girls believed that social restrictions hinder their education and careers.

> "With the onset of menstruation, most of the girls are confined to stay at home and discontinue our education."–(Female Adolescents: Slum-II, V & IX & School Teacher)

Boys, on the other hand, are allowed to continue their education while few from low-income families engage in productive work by assisting in the family business or working part-time.

Girls and boys engage in different recreational activities. Girls devote some time to self-care, including shopping, dressing up, grooming, visiting relatives, and watching TV. Boys spend the majority of their time playing sports and socializing with their peers outside home. Girls and their mothers reported unequivocally that boy groups verbally and physically harass girls while sitting at hotels or wandering on streets. This places girls at risk for sexual abuse and causes girls and their parents' stress.

> "While sitting in hotels or wandering on the streets, boy's groups harass girls verbally and physically, which sometimes results in sexual abuse."- (Mother: Slum II, Female Adolescents: Slum-III, VI, & All Stakeholders)

Even when probed, boys and fathers did not consider that boys' groups hanging out at small street hotels or wandering the streets could cause harassment for girls.

The ownership and use of mobile phones also differ between girls and boys; more boys own and use cell phones than girls. Mothers agreed that most boys and a few girls misuse their cellphones.

> "Boys most of the time are busy playing with their mobile phones; at times, this distracts them from studies and routine work."- (Mothers: Slum-VI & School Teacher)

### 4. Adolescents' social and health issues

All the participants agreed that adolescents still lack basic necessities such as safe drinking water, electricity, cooking gas, and security. It was further expressed that lack of these basic necessities impedes academic achievements and employment opportunities. Hence, it causes unnecessary mental stress during this period of physical and mental development.

> "Our children do not have basic facilities, and this is one of the barriers to their physical development and educational achievements."–*(Parents & Community Leader)*

According to stakeholders, there has been a proliferation of private educational entities over the last two decades; however, these facilities do not provide quality education.

"Many newly constructed schools do not provide quality education."–(*Madrassa & School Teacher*)

Parents and stakeholders informed that there are physical and sexual risks for adolescents, both boys and girls but more for girls.

"There are physical and sexual hazards for boys and girls, but more for girls."- (*"Mother: Slum-III & Stakeholder*)

Parents restrict girls' mobility and marry them immediately after puberty to protect them from physical and sexual abuse. Adolescents reported that mobility restrictions cause stress because these are imposed unexpectedly and without explanation. They also believed that before the age of 20, they are not prepared to handle in-laws and spousal relationships, including sexual encounters.

"Adolescents do not know anything about sex, sexual health, and marital relationships; they face difficulties in handling spousal relationships if married before age 20."–(*Adolescents: Slum-1I, V & VIII & Healthcare Providers*)

Adolescent boys also believed that early marriage imposes a financial burden on married boys.

"Early marriage adds an economic burden on boys as they have to fulfill their families' requirements."–(*Male Adolescents: Slum-1V, VI & X*)

All respondents agreed that there are no social or sports facilities for adolescent recreation. Moreover, they are unaware of good and bad behaviors and practices. As a result, boys engage in a variety of bad behaviors and practices such as smoking, tobacco use, alcohol, and drugs, as well as criminal activities like drug selling, theft, mobile snatching, and physical and sexual harassment. Few even join criminal gangs.

"Boys get involved in criminal activities and even join criminal gangs." (*Mother: Slum-V, NGO Representative, School Teacher*)

As a result of these risky behaviors, boys face friendship conflicts, injuries, road traffic accidents, and police arrests.

Adolescent girls and stakeholders reported a lack of understanding about sexual and reproductive health, as well as existing physical and sexual hazards. With increasing opportunities for interaction with boys through education and employment, many girls develop friendships with boys, which frequently leads to relationship conflict. As girl-boy friendship is still culturally unacceptable in this society, girls are unprepared to handle relationships with boys. Furthermore, adolescents keep these relationships secret as discussion about these relationships is considered a taboo subject.

"Girls develop relationships with boys secretly."–(*Female Adolescents: Slum-1, III, VII & XI*)

Due to a lack of knowledge and skills to deal with such relationships and absence of guidance, many girls are subjected to bullying, blackmailing, and physical and sexual violence that sometimes ends in murder or suicide.

"Girls are the victims of violence by boyfriends, which in extreme cases results in murder or suicide."–*(Female Adolescents: Slum-1, III, VII & XI, School & Madrassa Teachers and Healthcare providers)*

Despite being probed, parents did not acknowledge these girl-boy relationships, the resulting conflicts and unfortunate outcomes.

### 5. Adolescents' decision-making process

Parents and adolescents revealed that, according to cultural norms, adolescents could only participate in decisions about their education, employment, or minor day-to-day issues like clothing and food. All major decisions, especially those related to marriage, are taken by parents, specifically the father. Parents were convinced that adolescents are incapable of making decisions.

"It will be disastrous if adolescents start making decisions."—(*Mother: Slum-V, Political and Community Leaders)*

Most adolescents stated that they mostly discuss their problems with their friends and make decisions. Friends are especially preferred for discussing physical and sexual issues because adolescents believe that cultural norms prevent them from discussing these issues with their parents.

"Our cultural norms do not permit us to share our physical and sexual issues with parents."–(*Boy: Slum-I, V, VI & VIII)*

Another reason for preferring friends is that adolescents believe that both parents work excessively hard to manage the large family size and therefore have no time for adolescents. There is no communication with fathers because most of them are the family's sole breadwinners. As a result, they work longer hours and, in some cases, double shifts. This causes physical and mental exhaustion in fathers. As a result, they are usually in a bad mood and are unwilling to listen to the problems that adolescents face. Mothers are also busy with household chores, but they are reached out more often as they are welcoming.

### 6. Adolescents information needs, mechanism, and channels accessed

According to all the participants, social media networks such as Facebook, Instagram, and WhatsApp are the most frequently accessed information channels for both genders. In addition, they also get information from friends and television.

Parents, teachers and leaders were concerned that social media is disseminating inaccurate and socially unacceptable information, as well as undermining societal norms and value systems.

"Social media is spoiling the new generation."–(Parents, School and madrassa teachers, Political and Community leaders)

All respondents agreed that adolescents are in desperate need of information, particularly on pubertal physical and sexual changes, minor puberty-related ailments, existing physical, social, and environmental hazards, healthy behaviors and practices, communication skills, career opportunities, the importance of good company, the harmful effects of addictions, and

a correct version of an Islamic lifestyle, specifically concerning contraception. Adolescents should also be equipped with skills that will help them grow into responsible, healthy, and productive adults.

> "Adolescents should be provided with information on pubertal physical and sexual changes, minor puberty-related ailments, existing physical, social, and environmental hazards, healthy behaviors and practices, communication skills, career opportunities, the importance of good company, the harmful effects of addictions, and a correct version of an Islamic lifestyle, specifically concerning contraception. They should also be taught skills for handling the physical, social and environmental hazards and enabling them to grow into responsible, healthy, and productive adults."–*(All Male ^ Female Adolescents & Healthcare Providers)*

Adolescents believed that teachers should play a larger role in providing all this information because teachers and students have a friendlier relationship. However, for out-of-school adolescents, friends can play an active role.

## Discussion

This study aimed to understand the lives of adolescents from the perspectives of stakeholders in terms of their characteristics, knowledge about pubertal changes, roles and responsibilities assigned to them, health and social issues they face, their decision-making processes, and mechanisms and channels they use to gather information. This study significantly contributes to the existing literature by providing a comprehensive exploration of the multifaceted experiences of adolescents in squatter settlements of Karachi, Pakistan. Unlike previous research which often focused narrowly on specific aspects of adolescent life, our study offers a holistic view, integrating perspectives from adolescents, their parents, and key stakeholders. We have uncovered intricate details about gender-specific roles, decision-making processes, and the unique challenges faced by adolescents in these communities. Particularly, our findings regarding the influence of gender norms on adolescents' lives and the resultant disparities in education and social freedoms provide critical insights for policymakers and educators. Furthermore, the study sheds light on the pivotal role of communication gaps and the influence of social media, offering a nuanced understanding of the information channels utilized by adolescents. Such comprehensive insights are vital for developing targeted interventions and policies aimed at improving adolescent well-being in similar socio-cultural contexts (Refer to Fig 3).

Our study shows that all the participants were familiar with adolescent's characteristics and their activities. The study discovered that adolescent boys and girls are assigned different roles and responsibilities; girls are made responsible for household chores, whereas boys are assigned outside tasks. This finding is consistent with evidence found from other studies [25, 26]. The study also identified that mobility restrictions imposed on girls have an impact on their ability to continue their education resulting in dropouts. Similar findings have been found in studies from countries with comparable social backgrounds [27, 28]. Gender norms, which are powerful pervasive beliefs about gender-based social roles and practices that are deeply embedded in social structures, are the source of these restrictions [27]. These gender norms are created by patriarchal power dynamics and maintained, in part, through self-surveillance as girls follow internalized gender scripts. Literature suggests strategies for overcoming gender stereotyping and reducing its effects on gender and social role formation, such as promoting a positive role for media and stabilizing a gender-balanced domestic environment [29]. It is necessary to raise awareness about the negative effects of gender roles. Studies have

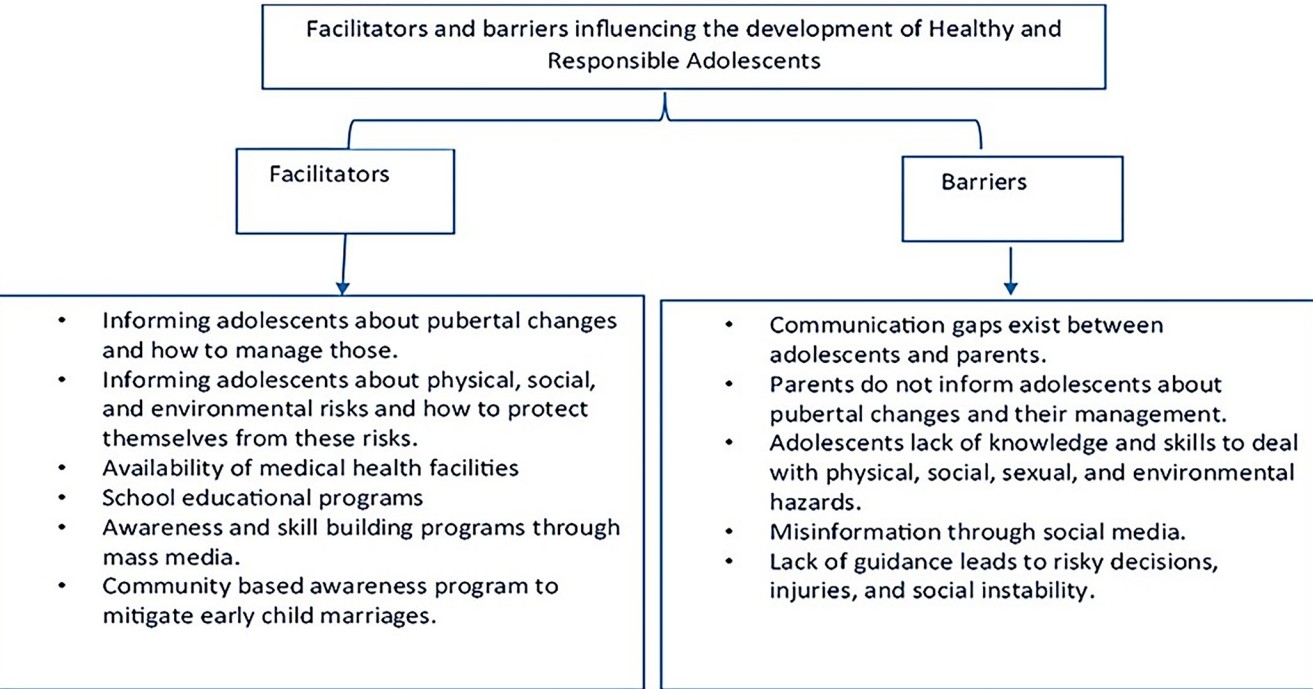

**Fig 3. Summary of facilitators and barriers that influencing the development of healthy and responsible adolescents.**

demonstrated that Behavior Change Communication (BCC) has changed gender norms [30]. Context-specific and tailored BCC interventions have the potential to overcome gender stereotyping and reduce its effects on gender roles, which can help to stabilize the gender-balanced domestic environment.

This study identified that a lack of necessities hinders the achievement of educational goals and, hence, career opportunities. Existing research also shows that poor socio-economic conditions have a negative impact on education and employment [31–35]. According to our study, adolescents are not adequately prepared to deal with existing physical, social, sexual, and environmental hazards. Moreover, communication gaps exist between adolescents and parents because discussing these risks is a taboo subject. Lack of knowledge and skills to deal with these challenges, as well as a lack of guidance, leads to risky decisions based on personal choices and peer advice. As a result of these poor decisions, adolescents face physical, social, sexual, and environmental issues, which results in injuries or even death. Studies show similar cases where adolescent girls are constantly subjected to bullying and sexual harassment [36–38] as a result of pubertal changes and aggression [36]. This emphasizes the importance of preparing adolescents to manage pubertal changes, take on new roles and responsibilities, and protect themselves from physical, social, and environmental risks. Studies have shown that parents, teachers, and healthcare providers are in the best position to communicate effectively with their children, listen to their needs, counsel, and provide guidance [39]. This clearly highlights the importance of raising awareness among parents, teachers, and healthcare providers about the risks that adolescents face, the significance of adolescent preparedness to protect themselves from these risks, and their role in this preparedness.

This study underscores the importance of acknowledging and empowering adolescents' agency, advocating for interventions that not only address the external constraints imposed by gender norms and socio-economic conditions but also harness the inherent capabilities of

adolescents to make informed decisions about their lives. Studies prove that adolescents can exhibit a dynamic role in navigating their health, education, and social interactions, contrary to the traditional view of them as passive participants in their developmental journey [40]. Our study showed that adolescents are obtaining information through easily accessible alternate pathways and mechanisms such as social media and television, as previously reported in other studies [40]. These information channels occasionally spread incorrect and socially unacceptable information that misleads adolescents. According to evidence, adolescents seek information to make informed decisions about their physical, sexual, and reproductive health, lifestyle, social relationships, communication, decision-making, and religion [41–49]. This insight calls for a paradigm shift in how we conceptualize and implement adolescent health and wellbeing programs, emphasizing a collaborative approach that values and leverages adolescents' inputs and perspective.

The current study findings, as well as previous research, highlight the importance of empowering adolescents to effectively manage physical, sexual, social, and environmental challenges. Research propose several strategies to help adolescents manage their challenges including high-quality health worker training, adolescent-responsive facilities, and broad community engagement have all been proposed [2, 50]. We propose that the government should establish youth-led platforms to co-create blended learning content on core and transferable skills with parents, teachers, and healthcare providers. The participation of social, political, community, and religious leaders is critical in promoting and disseminating this content to adolescents in informal education while also advocating for its inclusion in formal education.

## Conclusion

The study concludes that the adolescents face several physical, sexual, reproductive, social, and environmental challenges during the adolescent period. Adolescents are ill-equipped to carry out their new roles and responsibilities, effectively manage the challenges they face, and protect themselves from physical and sexual abuse. Because of social norms, there is a communication gap between adolescents and their parents. The most popular information channels are social media platforms and friends.

Based on our findings, we recommend specific actions for key stakeholders to support adolescents' development. Firstly, educators and policymakers should launch Behavior Change Communication (BCC) interventions aimed at dismantling gender stereotypes, thereby fostering a balanced domestic environment conducive to both boys' and girls' education and well-being. Secondly, it is imperative for parents, teachers, and healthcare providers to gain awareness about the multifaceted risks faced by adolescents, emphasizing their pivotal role in guiding and preparing the youth for these challenges. Lastly, we advocate for the creation of youth-led forums to collaboratively develop educational content alongside parents, educators, and health professionals, focusing on essential life skills. These efforts collectively can transform the demographic transition into a productive demographic dividend.

## Strengths and limitations of this study

This study stands out for its inclusive approach, incorporating a wide range of viewpoints from various stakeholders in the lives of adolescents across ten diverse areas of Karachi. This comprehensive perspective enriches the study's findings, offering a multifaceted understanding of adolescent experiences. The finding of this study provides comprehensive information about adolescents where there is a dearth of research on adolescents specially those living and urban slums in LMICs including Pakistan. Within this scenario this study provides evidence

to help policy makers and researchers to plan future interventions to improve adolescents' health and social well-being. The study's scope is confined to specific urban regions, impacting its generalizability. Future quantitative and qualitative studies can enhance our understanding of the breadth and depth of adolescent experiences in Pakistan and other LMICs.

## Declarations

### Ethical consideration & consent to participate

Ethical approval for this study was obtained from the Aga Khan University Ethical Review Committee (AKU-ERC). Written informed consent was provided by all study participants. Informed consent included permission to audio record the interviews and use anonymized quotes. Voluntary participation and the right to ask any questions and to decline participation at any time were emphasized during the data collection.

## Author Contributions

**Conceptualization:** Narjis Rizvi.

**Formal analysis:** Narjis Rizvi, Jawaria Mukhtar Ahmed.

**Investigation:** Narjis Rizvi, Sayyeda Ezra Reza, Rawshan Jabeen, Saleem Jessani.

**Methodology:** Sarah Saleem.

**Project administration:** Sayyeda Ezra Reza, Saleem Jessani.

**Resources:** Sarah Saleem, Jawaria Mukhtar Ahmed.

**Software:** Sarah Saleem, Jawaria Mukhtar Ahmed, Rawshan Jabeen.

**Supervision:** Narjis Rizvi, Sarah Saleem, Sayyeda Ezra Reza, Saleem Jessani.

**Validation:** Sarah Saleem, Jawaria Mukhtar Ahmed.

**Visualization:** Sarah Saleem.

**Writing – original draft:** Narjis Rizvi, Jawaria Mukhtar Ahmed, Saleem Jessani.

**Writing – review & editing:** Sarah Saleem, Sayyeda Ezra Reza, Rawshan Jabeen, Saleem Jessani.

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
