## [Decision Letter · Decision Letter 0]

29 Jan 2024

PGPH-D-23-02041

Are We Preparing Healthy & Responsible Adolescents? Exploratory Qualitative Study from Karachi, Pakistan

Dear Dr. Rizvi,

Thank you for submitting your manuscript to PLOS Global Public Health. After careful consideration, we feel that it has merit but does not fully meet PLOS Global Public Health’s publication criteria as it currently stands. Therefore, we invite you to submit a revised version of the manuscript that addresses the points raised during the review process.

The manuscript has been evaluated by two reviewers, and their comments are available below. The reviewers have raised a number of concerns that need attention. In particular they request additional information about the qualitative methods and analysis, and clarification about the contribution of this study in the context of the existing literature. Could you please revise the manuscript to carefully address the concerns raised?

We look forward to receiving your revised manuscript.

Kind regards,

Marianne Clemence

Staff Editor

Journal Requirements:

Additional Editor Comments (if provided):

Reviewers' comments:

Reviewer's Responses to Questions

**Comments to the Author**

1. Does this manuscript meet PLOS Global Public Health’s publication criteria? Is the manuscript technically sound, and do the data support the conclusions? The manuscript must describe methodologically and ethically rigorous research with conclusions that are appropriately drawn based on the data presented.

Reviewer #1: Yes

Reviewer #2: Yes

2. Has the statistical analysis been performed appropriately and rigorously?

Reviewer #1: N/A

Reviewer #2: N/A

3. Have the authors made all data underlying the findings in their manuscript fully available (please refer to the Data Availability Statement at the start of the manuscript PDF file)?

Reviewer #1: Yes

Reviewer #2: Yes

4. Is the manuscript presented in an intelligible fashion and written in standard English?

Reviewer #1: Yes

Reviewer #2: Yes

5. Review Comments to the Author

Reviewer #1: Thank you for the opportunity to review such an interesting piece of work. The article is well written, easy to read and comprehend. The single communication objective is clear. The study addresses a critical conversation that has significant implications for adolescent health and wellbeing policy and practice. The data speak to the conclusions sufficiently. Below are a few observations and recommendations;

Discussion: The paper could benefit from a stronger discussion section beyond confirming what other studies have found . What is the specific value add of this data? What are the key urgent messages? In addition, the paper could benefit from a theoretical basis. It is not clear what theoretical lens the authors are utilizing to shape the interpretation of findings and discussion. Grounding the discussion in theory will further strengthen the paper. The concept of adolescents' agency seems to be emerging but not very well discussed in the paper. Expounding on this could add much value to the paper, acknowledging the role of adolescents in their own health and wellbeing Vs positioning them as passive beings in their wellbeing and development.

Methods Section: The authors could add some more helpful detail to this section for instance, how many adolescents were interviewed? How many were male, female? How many FGDs were held? How many adolescents were in each FGD, what was the breakdown in terms of demographic characteristics beyond age? What languages were used in the discussions? Were there any particular play based or other specific approaches used in the discussions? All this would help the reader understand the setting of the FGDs better. Additionally, more description of the context would be helpful.

Conclusion: There is need for some specificity on who the recommendations are for. Who should do what? They are vey general. Highlighting potential areas for future research might also be useful.

Reviewer #2: Thank you for the opportunity to review the manuscript title “Are We Preparing Healthy & Responsible Adolescents? Exploratory Qualitative Study from Karachi, Pakistan”. Overall, the study focus is too broad: gaining the perspectives from teenagers, parents, and stakeholders regarding adolescents’ characteristics, activities they are involved in, health and social issues they face, their decision-making process, and mechanism and channels they use to gather information. This broad focus appears to have prevented the author from providing a more detailed, nuanced account of the topics included in the study objective. Bellow are specific comments.

Abstract

- Please specify the qualitative method/methodology that guided the study design. For example, qualitative description.

- In addition to a broad research focus, the authors explored the perspectives of adolescents, their parents, and “key adolescent stakeholders. Given the number of interviews and focus groups conducted, I wonder whether these perspectives that include seral issues need to be reported in separate papers.

- Use a language that suggests that authors are reporting the perspectives of the three groups rather that taking from granted what participants stated. For example: “Based on gender norms, parents assign specific roles and responsibilities to adolescent girls and boys”

- Abstract conclusion does not provide the key practical and research implications that can be derived from the study conducted. Instead, the authors highlight some remedial actions assumed to be affective to deal with the identified issues.

Introduction

- A reference is needed for the claim made in the third sentence of the first paragraph.

- Overall, this section is well-structured

Methods

- Like quantitative research, qualitative research is a research inquiry. Thus, the specific qualitative research method that guided the study design, if any, should be indicated and justified.

- Please specify the purposeful sampling strategy employed. What information was given to participants? Were they informed about their rights and roles?

- Apart from piloting the interview guides, were they informed by guidelines/approaches relate to developing semi-structured interview guides?

- Why did the authors claim that data collection continued until achieving theoretical saturation if this study does not aim to develop a substantive theory based on qualitative data?

- A heading should be used for data collection and for data analysis

- A reference is needed for the thematic analysis approach employed in the study. Further details are needed about the analysis performed.

- Rigor has to be demonstrated rather than suggested. Authors are supposed to state how the criteria of rigor were implemented.

Results

- Table 2 does not appear to provide characteristics of participants, but the total number of participants in each group. Highlighting these characteristics is important in qualitative research to enhance applicability/transferability of study findings, which was a criterion of rigor listed at the end of the previous section.

- Data analysis does not appear to have yielded themes but categories or topics within which the relevant data were sorted.

- Based on the study objective, it seems that the "themes" (topics of interest) were not derived inductively from the data but were predefined. This approach does not align with inductive thematic analysis as claimed in the previous section. If so, background information was not provided to support the need for exploring participants’ views of the defining characteristics of adolescence.

- In qualitative research, researchers are encouraged to report the diversity of views within and across groups involved in the study. In this study, it seems that authors only reported the views that participants across groups (e.g., adolescents, parents) shared.

- Quotes are needed to support the last interpretation (final paragraph) in this section.

- Overall, the analysis was appropriate but lacked detail and nuance due to the board study focus.

Discussion

- Please highlight in this section (preferably in the first paragraph) the study contribution, what it added to the current literature.

- Please add study limitations and future directions.

- Make the conclusions more concise.

6. PLOS authors have the option to publish the peer review history of their article (what does this mean?). If published, this will include your full peer review and any attached files.

**Do you want your identity to be public for this peer review?** For information about this choice, including consent withdrawal, please see our Privacy Policy.

Reviewer #1: No

Reviewer #2: No

---

## [Decision Letter · Decision Letter 1]

8 May 2024

PGPH-D-23-02041R1

Are We Preparing Healthy & Responsible Adolescents? Exploratory Qualitative Study from Karachi, Pakistan

Dear Dr. Rizvi,

Thank you for submitting your  revised manuscript to PLOS Global Public Health. After careful consideration, we feel that it has merit but does not fully meet PLOS Global Public Health’s publication criteria as it currently stands. Therefore, we invite you to submit a revised version of the manuscript that addresses the points raised during the review process.

Your manuscript has been assessed by 2 reviewers and their comments are available below. They feel the manuscript would benefit from a thorough explanation of the methodology and analysis. Could you carefully revise the manuscript to address their comments? 

We look forward to receiving your revised manuscript.

Kind regards,

Annesha Sil, Ph.D.

Staff Editor, PLOS 

Journal Requirements:

If you did not receive any funding for this study, please simply state: “The authors received no specific funding for this work.

2. We have noticed that you have uploaded Supporting Information files, but you have not included a list of legends. Please add a full list of legends for your Supporting Information files after the references list.

Additional Editor Comments (if provided):

Reviewers' comments:

Reviewer's Responses to Questions

**Comments to the Author**

1. If the authors have adequately addressed your comments raised in a previous round of review and you feel that this manuscript is now acceptable for publication, you may indicate that here to bypass the “Comments to the Author” section, enter your conflict of interest statement in the “Confidential to Editor” section, and submit your "Accept" recommendation.

Reviewer #1: All comments have been addressed

Reviewer #3: All comments have been addressed

2. Does this manuscript meet PLOS Global Public Health’s publication criteria? Is the manuscript technically sound, and do the data support the conclusions? The manuscript must describe methodologically and ethically rigorous research with conclusions that are appropriately drawn based on the data presented.

Reviewer #1: Yes

Reviewer #3: Yes

3. Has the statistical analysis been performed appropriately and rigorously?

Reviewer #1: Yes

Reviewer #3: Yes

4. Have the authors made all data underlying the findings in their manuscript fully available (please refer to the Data Availability Statement at the start of the manuscript PDF file)?

Reviewer #1: Yes

Reviewer #3: Yes

5. Is the manuscript presented in an intelligible fashion and written in standard English?

Reviewer #1: Yes

Reviewer #3: Yes

6. Review Comments to the Author

Reviewer #1: (No Response)

Reviewer #3: 1. The phrase in the title ought to be broad and indicate your objectives. Reframe in light of your themes, research design, and study area (line no: 1).

2. MeSH based keywords need to add (line no: 60).

3. The last paragraph of the introduction should include a mention of the problem statement or study scope. The goals of this study should be added rather than being the aim (line no: 109-113).

4. Create a separate statement for the study's purpose and design (line no: 116-119).

5. Rather than using take, describe the steps for choosing samples here. I suggest doing away with the tables. (line no: 124-134).

6. For the following data collection process, any alterations or reflections need to be noted in writing (line no: 139).

7. The data was gathered earlier than seven years. Do you believe the information is relevant to the current situation? because many modifications have occurred since COVID. How are you going to defend it? (line no: 145).

8. There needs to be a focus group and interview checklist. Here's an illustration of a COREQ checklist. When collecting data, were there guidelines that you followed? (line no: 135).

9. Please describe it (line no: 154).

10. The amount of data is enormous. Have you used any programme for theme analysis? utilised software such as Atlas TI, to name a few. Please mention that (Manuscrpit line no: 156).

11. My suggested order for the results is as follows: 1. Respondents' sociodemographic information, 2. The frequency of themes and subthemes (that is, the number of responses to each specific theme), 3. Themes with an objective and their subthemes, 4. The verbatims of the FGDs and IDIs Key Informant participants must be included under each subtheme. There must be at least two sentence interpretations and at least four to five verbatims under each subtheme, 5. Repossess number need to refer. For instance, (Male adolescent, Age, Number of Respondents) (line no: 171).

12. Parentage does not begin a sentence. I would advise rephrasing the sentence (line no: 173-174).

13. Not within the theme It ought to be explained in a table under the section titled "Respondents' sociodemographic details."(line no: 182).

14. The opinions of the respondents differed on this. Need to change it (line no: 195).

15. Please reframe according to the results (line no: 310).

16. Lacking directions for the future in the figure, and it ought to be addressed in light of current research (line no: 381-382).

17. Rephrasing limitations is necessary (line no: 400).

7. PLOS authors have the option to publish the peer review history of their article (what does this mean?). If published, this will include your full peer review and any attached files.

**Do you want your identity to be public for this peer review?** For information about this choice, including consent withdrawal, please see our Privacy Policy.

Reviewer #1: **Yes: **Clare Ahabwe Bangirana

Reviewer #3: No

---

## [Decision Letter · Decision Letter 2]

25 Jul 2024

PGPH-D-23-02041R2

Are We Preparing Healthy & Responsible Adolescents? Exploratory Qualitative Study to understand the health and social issues of adolescent living in Karachi, Pakistan

Dear Dr. Rizvi,

Thank you for submitting your manuscript to PLOS Global Public Health. After careful consideration, we feel that it has merit but does not fully meet PLOS Global Public Health’s publication criteria as it currently stands. Therefore, we invite you to submit a revised version of the manuscript that addresses the points raised during the review process.

The revised manuscript has been reviewed and reviewer 3 has requested a minor revision to the manuscript. Please refer to Appenidx 6 within the main text of the manuscript, as requested by reviewer 3.

We look forward to receiving your revised manuscript.

Kind regards,

Emma Campbell, Ph.D

Staff Editor

Journal Requirements:

Reviewers' comments:

Reviewer's Responses to Questions

**Comments to the Author**

1. If the authors have adequately addressed your comments raised in a previous round of review and you feel that this manuscript is now acceptable for publication, you may indicate that here to bypass the “Comments to the Author” section, enter your conflict of interest statement in the “Confidential to Editor” section, and submit your "Accept" recommendation.

Reviewer #3: All comments have been addressed

2. Does this manuscript meet PLOS Global Public Health’s publication criteria? Is the manuscript technically sound, and do the data support the conclusions? The manuscript must describe methodologically and ethically rigorous research with conclusions that are appropriately drawn based on the data presented.

Reviewer #3: Yes

3. Has the statistical analysis been performed appropriately and rigorously?

Reviewer #3: Yes

4. Have the authors made all data underlying the findings in their manuscript fully available (please refer to the Data Availability Statement at the start of the manuscript PDF file)?

Reviewer #3: Yes

5. Is the manuscript presented in an intelligible fashion and written in standard English?

Reviewer #3: Yes

6. Review Comments to the Author

Reviewer #3: Appenidx 6 should be address in the as a text reference.

7. PLOS authors have the option to publish the peer review history of their article (what does this mean?). If published, this will include your full peer review and any attached files.

**Do you want your identity to be public for this peer review?** For information about this choice, including consent withdrawal, please see our Privacy Policy.

Reviewer #3: No

---

## [Editor Report · Decision Letter 3]

29 Aug 2024

Are We Preparing Healthy & Responsible Adolescents? Exploratory Qualitative Study to understand the health and social issues of adolescent living in Karachi, Pakistan

PGPH-D-23-02041R3

Dear Dr. Rizvi,

We are pleased to inform you that your manuscript 'Are We Preparing Healthy & Responsible Adolescents? Exploratory Qualitative Study to understand the health and social issues of adolescent living in Karachi, Pakistan' has been provisionally accepted for publication in PLOS Global Public Health.

Best regards,

Julia Robinson

Executive Editor